# Engineering Optogenetic Control of Endogenous p53 Protein Levels

**Pierre Wehler** [1,2] **and Barbara Di Ventura** [1,2,*] 

[1]  Faculty of Biology, University of Freiburg, 79104 Freiburg, Germany; pierre.wehler@biologie.uni-freiburg.de
[2]  Signalling Research Centres BIOSS and CIBSS, University of Freiburg, 79104 Freiburg, Germany
*   Correspondence: barbara.diventura@biologie.uni-freiburg.de

**Featured Application: This study describes the development of an optogenetic tool to control endogenous p53 levels with blue light.**

**Abstract:** The transcription factor p53 is a stress sensor that turns specific sets of genes on to allow the cell to respond to the stress depending on its severity and type. p53 is classified as tumor suppressor because its function is to maintain genome integrity promoting cell cycle arrest, apoptosis, or senescence to avoid proliferation of cells with damaged DNA. While in many human cancers the p53 gene is itself mutated, there are some in which the dysfunction of the p53 pathway is caused by the overexpression of negative regulators of p53, such as Mdm2, that keep it at low levels at all times. Here we develop an optogenetic approach to control endogenous p53 levels with blue light. Specifically, we control the nuclear localization of the Mmd2-binding PMI peptide using the light-inducible export system LEXY. In the dark, the PMI-LEXY fusion is nuclear and binds to Mdm2, consenting to p53 to accumulate and transcribe the target gene p21. Blue light exposure leads to the export of the PMI-LEXY fusion into the cytosol, thereby Mdm2 is able to degrade p53 as in the absence of the peptide. This approach may be useful to study the effect of localized p53 activation within a tissue or organ.

**Keywords:** Optogenetics; p53; *As*LOV2; LINuS; LEXY; MIP; PMI

## 1. Introduction

Despite its unassuming name, p53 is an essential protein for the cell and is evolutionarily conserved from worm to human [1,2]. Acting as a sensor of a variety of stress signals, p53 dictates life or death to the cell, depending on whether it is able or not to cope with the stress and return to homeostasis [3]. The fundamental role of p53 in ensuring genomic integrity is better reflected in its nickname, "guardian of the genome". This and other epithets, such as "good cop/bad cop" or "heavily dictated dictator", emphasize not only its centrality in keeping proliferating cells under control, but also suggest that p53 can have contradicting effects on the organism (like cancer prevention and aging) and is therefore itself subjected to a tight regulation achieved by a complex cellular network [4,5]. p53 is mainly a sequence-specific DNA-binding protein which, under normal growth conditions, is kept latent in various ways [5,6]. The most prominent mechanism to keep p53 latent when not needed is to maintain it at very low levels. This is achieved by a series of E3 ubiquitin ligases that target p53 for proteasomal degradation by ubiquitylating it on lysine residues [7,8]. The oncoprotein murine double minute 2 (Mdm2) appears to be the major negative regulator of p53 [9,10]. Beyond targeting it for degradation, Mdm2 inhibits p53 also by promoting its nuclear export [9] and by binding to its N-terminal transactivation domain (TAD), which is therefore not available for binding to the DNA [11–13]. Interestingly, the Mdm2 homologous protein, murine double minute 4 (MdmX), inhibits

p53 via the same TAD-sequestration strategy, but lacks the ability to promote p53 degradation, despite having a RING domain like Mdm2 [14].

When necessity calls, i.e., upon stress signals such as DNA damage, hypoxia, or deregulated expression of oncogenes, p53 is rapidly activated via acquisition of post-translational modifications, some of which enable its stabilization and the consequent increase in its protein levels [5]. This post-translationally modified, stabilized p53 can then start its gene expression program. p53 target genes mediate a plethora of cellular functions, including cell cycle arrest, senescence, apoptosis, differentiation, and DNA repair, to name but a few [2].

Inactivation of the p53 pathway—be it through mutations of the p53 gene itself, mutations of genes encoding key p53 regulators, and/or binding partners, or other means—is a common, if not universal feature of human cancers. The cancers in which p53 is not mutated, but rather aberrantly kept at low levels at all times by means of overexpressed negative regulators [15,16], have better chances of being defeated because, if p53 levels could be brought back to normal, its target genes could be activated. Indeed, reactivation of the endogenous p53 signaling pathway has been shown to lead to tumor regression [17,18]. Therefore, disrupting an aberrant p53-Mdm2/MdmX interaction on a wild type p53 background could be a promising therapy.

Another feature that would make a new therapy promising would be the ability to target it only at the cancer cells. Indeed, the major drawback of conventional chemotherapy is the lack of specific distribution of the drugs to the interested tissues and organs. The drugs distribute non-specifically inside the body, leading to substantial damage of healthy cells alongside the diseased ones. To overcome this problem, new concepts have been developed, such as delivery of chemotherapeutic drugs from vehicles (nanoparticles, liposomes, polymeric micells, minicells, etc.) that accumulate inside cancer cells for instance via ligand-receptor interactions [19] or that release their content locally upon illumination with near-infrared (NIR) light [20,21].

Alongside its use in combination with chemical compounds, light has been exploited to regulate genetically encoded photosensors within cells, giving rise to the field of optogenetics [22,23]. One of the workhorses of optogenetics is the second light-oxygen-voltage (LOV) domain of *Avena sativa* phototrophin 1 (*As*LOV2 domain). In the dark, the C-terminal Jα helix of the *As*LOV2 domain is folded and bound to the core domain [24–28] once the chromophore flavin mononucleotide (FMN) absorbs photons of ~450 nm light, a structural rearrangement occurs within the protein triggering the unfolding and undocking of the Jα helix [24–28]. The recovery to the inactive dark state occurs spontaneously within less than a minute [29]. This conformational change has been harnessed to control, with light, the exposure of diverse peptides that have been appended to the Jα helix [30–32]. In particular, nuclear localization and export signals (NLSs and NESs) were photocaged within *As*LOV2 giving rise to the optogenetic tools named LINuS [33] and LANS [34] for controlling nuclear protein import, and LEXY [35] and LINX [36] for controlling nuclear protein export.

Here we explore the possibility to use optogenetics to elevate p53 levels by disrupting its interaction with Mdm2 and MdmX. We compare two previously published peptides, MIP [37] and PMI [38], and find that, under our experimental conditions, only PMI results in higher p53 protein levels in HCT116 cells. Interestingly, we discover that the peptide needs to localize to the nucleus to exert its function. Therefore, we fuse it to LEXY and show that the PMI-mCherry-LEXY construct allows controlling with light endogenous p53 levels in HCT116 cells. Specifically, cells kept in the dark have higher p53 levels than cells exposed to blue light which triggers export of the PMI-mCherry-LEXY construct into the cytosol, thereby preventing PMI to exert its function. Stabilized p53 leads to the accumulation of the p53 target gene p21 [39,40] in HCT116 cells kept in the dark. Even if not applicable in a therapeutic scenario in the present form—as cells with elevated p53 levels are, in this case, those left in the dark—this approach should be useful to study the effect of having a heterogeneous p53 status within a tissue or organ.

## 2. Materials and Methods

### 2.1. Plasmid Generation

The plasmids expressing either p53 or thioredoxin bearing the MIP/PMI/3A peptide fused to mCherry were constructed from pcDNA3.1(+) (ThermoFischer Scientific) by first inserting the mCherry coding sequence between the BamHI and XhoI sites of the multi-cloning site (MCS) of pcDNA3.1(+) and then inserting the p53 or thioredoxin coding sequence into this modified vector between the HindIII and BamHI sites. The NLS/NES upstream of mCherry was cloned by adding the corresponding sequence directly into the forward primer used to amplify the *mcherry* gene. Peptides were inserted as oligos into thioredoxin between Gly33 and Pro34 using golden gate cloning. Addition of PMI-linker(GGS)-NLS sequences to mCherry, LINuS, or LEXY constructs was done using overhang PCR. All constructs were verified by DNA sequencing. The LINuS and LEXY constructs used here have been previously described [33,35].

### 2.2. Illumination of Cells with Blue Light

Cells were illuminated with 20 $\mu$mol s$^{-1}$ m$^{-2}$ of 480 nm light within the cell culture incubator using a custom-made LED device connected to a power box (Manson HCS-3102) controlled by a Raspberry Pi.

### 2.3. Cell Culture and Transient Transfection

The human colon carcinoma cell line HCT116 was kindly provided by Thomas Hofmann, University of Mainz. HCT116 cells were maintained in phenol red-free Dulbecco's Modified Eagle Medium supplemented with 10% FCS (Sigma), 2 mM L-glutamine (Life Technologies) and 1% PenStrep (Life Technologies). Cells were cultured at 37 °C and 5% $CO_2$ in a humidified tissue culture incubator. Cells were transfected with 2500 ng total DNA (250 ng construct DNA and 2250 ng empty pcDNA3.1 (+)) using Lipofectamin 2000 according to the manufactures protocol. Mock transfected samples were transfected with empty pcDNA3.1 (+).

### 2.4. Western Blot

Cells were seeded into 6-well plates and transfected as described above. For non-optogenetics experiments, cells were kept in the dark at all times and lysed 32 h post-transfection. For optogenetics experiments, cells were incubated right after transfection under blue light (20 $\mu$mol s$^{-1}$ m$^{-2}$; 480 nm) overnight. The next day, one plate was incubated in the dark (dark control), while the other plate(s) remained under illumination. 24 h later, cells were collected in ice-cold lysis buffer (20 mM Tris-HCl pH 7.4, 1% Triton X.100, 10% glycerol, 150 mM NaCl, 1% phenylmethylsulfonyl fluoride, 1% benzonase (Novagen, 70664), and 1 Complete Mini Protease Inhibitor tablet (Roche, 11 836 153 001)). In the experiment shown in Figure 1b,c, the double-strand DNA breaks-inducing anthracycline antibiotic daunorubicin was added at a final concentration of 0.5 $\mu$M 8 h after transfection. Protein concentration was measured by Bradford assay and adjusted to 1 $\mu$g\mL. 15 $\mu$g were loaded on a 12% Bis-Tris gel and proteins were separated by electrophoresis. Proteins were then transferred onto a polyvinylidene difluoride membrane, which was blocked using 5% BSA in PBS-Tween 20 (PBS-T). Primary antibodies were diluted in 5% BSA in PBS-T and applied for 1 h to detect p53 (Santa Cruz, sc-126, diluted 1:1000) or p21 (BD Pharmigen, 556430, diluted 1:666) and beta-actin (Abcam, ab8226, diluted 1:1000), followed by incubation with a secondary goat anti-mouse IgG(H+L)-PRP0 (Dianova, 115-035-003) for 45 min. Chemiluminescence was detected using the SuperSignal West Pico Chemiluminescent Substrate (Thermo Scientific) and the ChemoCam Imager (Intas).

## 2.5. Confocal Imaging

Cells were imaged at 37 °C and 5% $CO_2$ in ibidi μ-slide 8 wells (ibidi, 8226) on a Leica Sp5 confocal microscope equipped with an incubation chamber and a 20x air objective (0.7 NA). Activation was performed imaging cells with the 458 nm laser at 80% intensity for 5 seconds every 30 s for a total of 40 min. mCherry was excited using the 561 nm laser. Cells were focused with the 561 nm laser to prevent premature activation of LINuS/LEXY.

## 2.6. Data Processing

### 2.6.1. Quantification of Protein Levels

Western blots were quantified by measuring band intensities using the ImageJ Gels package. The intensity of the band corresponding to the loading control was used to normalize the intensity of the band corresponding to the protein under investigation (p53 or p21).

### 2.6.2. Image Analysis

Microscopy images were processed using ImageJ. First, the background was subtracted, then a circular ROI was drawn manually in the nuclei of all cells in a given field of view. The mean intensity of the nuclei was quantified at each time point. Values are normalized to that at time point zero (before illumination started).

### 2.6.3. Statistical Analysis

Independent replicates refer to independent cell samples of experiments carried out on different days. If not indicated otherwise, bars and graph represent the mean values, the error bars the standard deviation of the mean. Statistical analysis was done by a two-tailed unpaired Student's *t*-test; P values (p) < 0.05 (*), < 0.01 (**), and < 0.001 (***) were considered statistically significant. P values ≥ 0.05 were considered statistically not significant (NS).

## 3. Results

### 3.1. The Mdm2-inhibitory Peptide (MIP) Does Not Elevate p53 Levels in HCT116 Cells

We started by testing the Mdm2-inhibitory peptide MIP (PRFWEYWLRLME), which was selected *in vitro* from large libraries of random peptides using mRNA display [37]. MIP binds Mdm2 and also MdmX with higher affinity than other known peptides, such as DI [41]. We cloned the MIP peptide into a freely accessible loop of *Escherichia coli* thioredoxin, a small and stable protein commonly used as scaffold for presenting and stabilizing peptides. mCherry was added to the construct to consent easy determination of transfection efficiency.

As a control, we cloned the 3A peptide (LTAEHYAAQATS) in which three key hydrophobic residues of the DI peptide are mutated to alanine, thereby abrogating binding to Mdm2 or MdmX.

To test the effect of the peptide we selected HCT116 cells, a human colon cancer cell line with wild type p53, which has been shown to accumulate upon inhibition of Mdm2 [42]. Expression of Trx-MIP-mCherry did not lead to changes in p53 protein levels, which remained similar to those in mock and Trx-3A-mCherry transfected samples (Supplementary Figure S1). This is in contrast with previously published results [37]. The raw data used to quantify protein levels are found in the Supplementary Materials (Figures S2–S5).

### 3.2. The p53-Mdm2/MdmX Inhibitor (PMI) Peptide Elevates p53 Levels in HCT116 Cells

We then turned to another peptide, the so-called p53-Mdm2/MdmX inhibitor (PMI) peptide [38] (TSFAEYWNLLSP). PMI was identified in a phage display, using as baits the p53-binding domains of Mdm2 and MdmX. We followed the same approach as for the MIP peptide, thus cloning PMI into the thioredoxin scaffold fused to mCherry, and then testing its function in HCT116 cells.

36 h after transfection, cells expressing Trx-PMI-mCherry exhibited about 60% increase in p53 levels compared to the mock transfected cells, implying that degradation of p53 was inhibited (Figure 1a). As seen in previous experiments, expression of Trx-3A-mCherry and Trx-MIP-mCherry did not affect p53 levels.

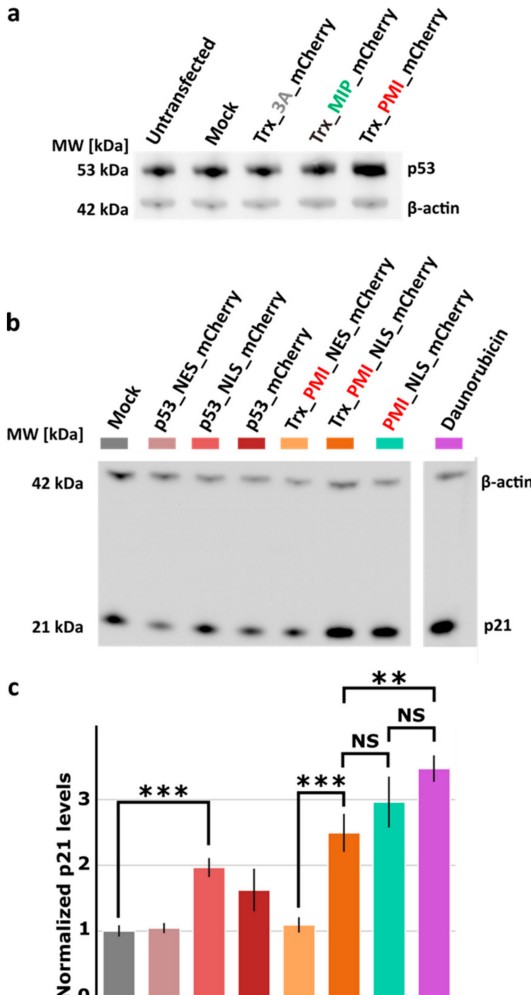

**Figure 1. The p53-MDM2/MDMX inhibitor (PMI) peptide stabilizes p53 in HCT116 cells in a localization-dependent manner.** (a), (b) Western blot showing the levels of p53 (a) and p21 (b) in HCT116 cells transfected with the indicated constructs. Mock indicates cells transfected with empty pcDNA3.1(+). β-actin was used as loading control. (**a**) This experiment was conducted once. (**b**) Daunorubicin was added at 0.5 µM 8 h after transfection. Cells were collected for Western blot analysis after 24 h. A lane with a sample not relevant to this study was cut from the image and is shown as a white space. (**c**) Bar plot showing the quantification of two independent Western blots with samples as in (b). Data indicate mean ± standard deviation. Color code as in (b). NS p ≥ 0.05, ** p < 0.01, *** p < 0.001, *t*-test.

Next, we asked whether PMI had to localize to either the nucleus or the cytosol to exert its function. For this purpose, we added a nuclear localization signal (NLS) or a nuclear export signal (NES) to the Trx-PMI-mCherry construct. This time, we decided to look directly at the effect of the peptide on one of p53 target genes, p21. Indeed, it is important to make sure that stabilized p53 is fully functional. As additional controls beyond the mock transfected cells, we included cytoplasmic p53 (p53-NES-mCherry), nuclear p53 (p53-NLS-mCherry and p53-mCherry, which is largely nuclear even

in the absence of an additional NLS) and daunorubicin, a double-strand DNA breaks-inducing drug. Interestingly, we found that the peptide had to be nuclear to elevate p21 levels (Figure 1b,c).

We then tested if we could dispense of the thioredoxin scaffold by expressing PMI simply as a fusion to mCherry. This would further reduce the size of the final construct. We found that the PMI-mCherry fusion worked as well as the construct based on thioredoxin, giving rise to p53 levels similar to those obtained when using daunorubicin (Figure 1b,c).

### 3.3. Developing Optogenetic Control of PMI Localization

As the function of PMI is dependent on its nuclear localization, we reasoned that, if we could control this latter with LINuS, we could effectively control endogenous p53 levels with blue light. To this aim, we cloned three constructs differing only in the LINuS variant used, as different NLSs photocaged in the Jα helix of the *As*LOV2 domain have different properties such as background in the dark or extent of nuclear accumulation after light induction [33] (Figure 2a). Albeit two of the three constructs showed statistically significant nuclear accumulation after light induction (Figure 2b,e), we were not satisfied with their performance, as the fold change in nuclear localization before and after light was less than two.

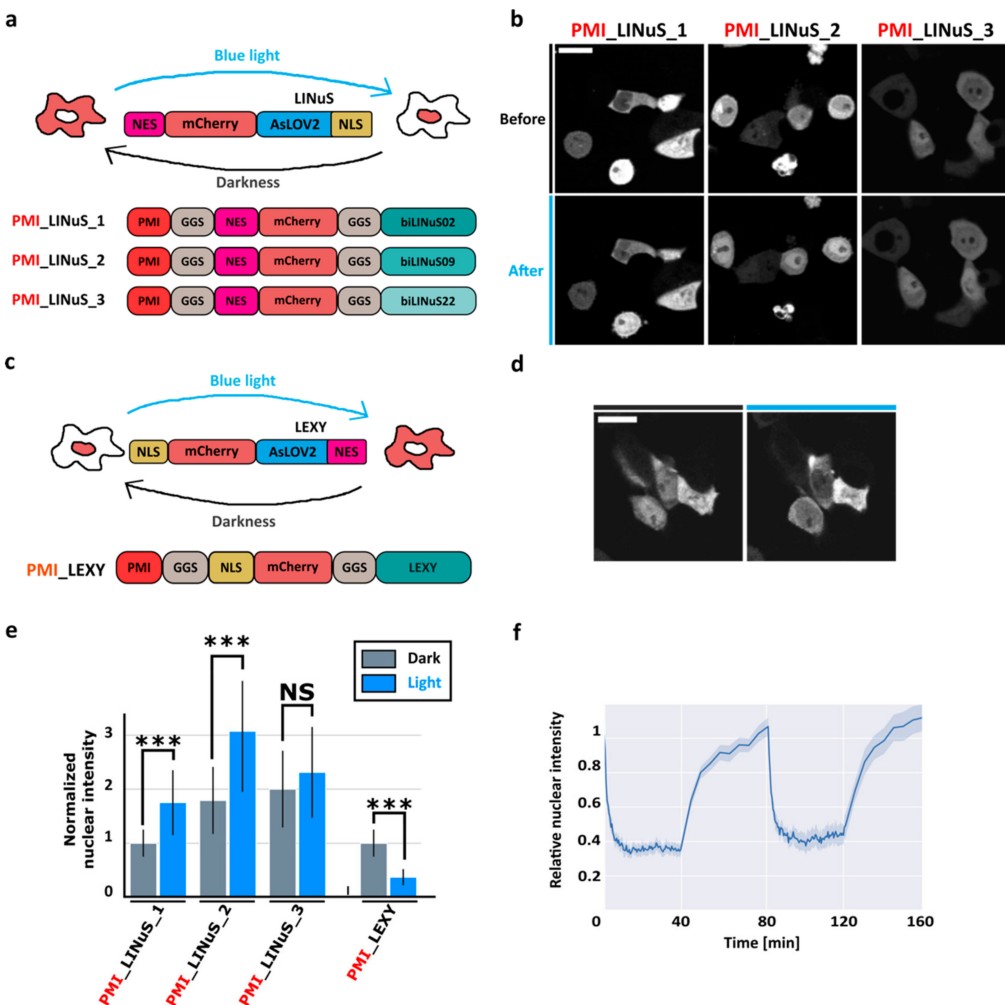

**Figure 2. Engineering optogenetic control of PMI localization.** (**a**,**c**) Schematic representation of the PMI_LINuS and PMI_LEXY constructs. The NES is the PKIt NES. GGS is a flexible linker. (**b**,**d**) Representative images of HCT116 cells transiently transfected with the indicated constructs before and after 40 min of blue light illumination. (**b**,**d**) Scale bar, 20 μm. (**e**) Bar plot showing the nuclear intensity of the indicated constructs in the dark or after illumination with blue light normalized to

the values for the PMI_LINuS_1 construct in the dark (for the LINuS constructs) and to PMI_LEXY construct in the dark (for the LEXY construct). NS p ≥ 0.05, *** p < 0.001, *t*-test. (**f**) Plot showing the nuclear intensity of the mCherry fluorescence measured in HCT116 cells transiently transfected with the PMI_LEXY construct shown in (**c**) in cells illuminated with blue light twice with a 40 min-long activation phase. Graph represents mean plus 95% confidence interval. N = 23 cells.

Therefore, we turned to LEXY, which is similar to LINuS in that it is also based on *As*LOV2, however, in this case, the PMI peptide would be nuclear unless light is applied to the cells. LEXY has proven to be easily applicable to various proteins of interest without the need of optimization [35,43,44]. PMI fused to mCherry and LEXY (PMI-NLS-mCherry-LEXY, for simplicity referred to as PMI_LEXY from now on) could be robustly accumulated into the cytoplasm after blue light illumination of HCT116 cells (Figure 2c–e), showing a fold change close to three. We further confirmed the reversibility of this accumulation (Figure 2f).

### 3.4. PMI_LEXY Can Be Used to Control with Light Endogenous p53 and p21 Levels

Next, we asked whether we could dictate p53 levels by externally applying blue light to the cells expressing PMI_LEXY. We found that cells kept in the dark had ~3-fold higher p53 levels than mock transfected or illuminated cells (Figure 3a,c). Finally, we assessed whether we could also detect higher p21 levels in HCT116 cells kept in the dark. This was indeed the case (Figure 3b,c).

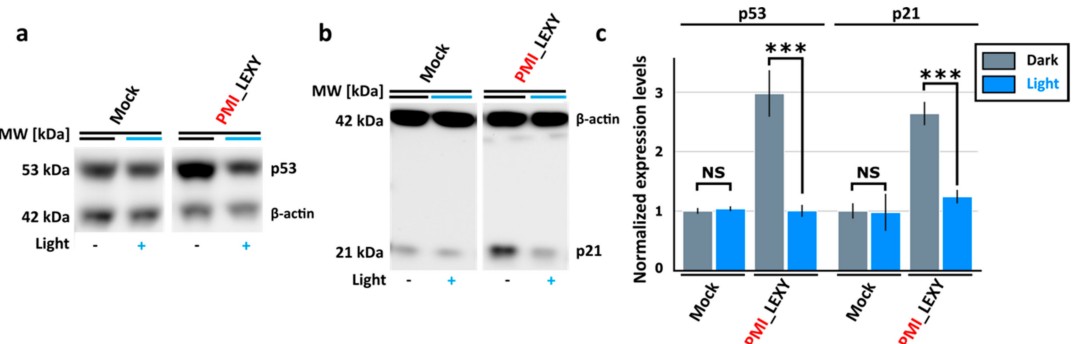

**Figure 3. LEXY allows controlling with light endogenous p53 and p21 levels.** (a, b) Western blot showing the levels of p53 (**a**) and p21 (**b**) in HCT166 cells. Mock indicates cells transfected with empty pcDNA3.1(+). β-actin was used as loading control. (**c**). Bar plot showing the quantification of three independent western blots with samples as in (a) and (b). Levels are shown normalized to the levels of the mock, dark control. NS p ≥ 0.05, *** p < 0.001, *t*-test.

## 4. Discussion

p53 protein levels are mainly regulated by the E3 ubiquitin ligase Mdm2, which triggers p53 ubiquitylation and consequent proteasomal degradation. Mdm2 additionally inhibits p53 by binding to its N-terminal transactivation domain (TAD), which is thus not free for contacting the transcription machinery. This second mechanism of p53 inhibition is shared by other proteins, such as MdmX, which do not lead to p53 degradation.

As some tumors are characterized by wild type p53 but aberrant overexpression/activity of one or more of its negative regulators, a promising strategy to combat them is to restore normal p53 signaling by stopping the aberrant inhibition the protein is subject to.

Peptides that bind to Mdm2/MdmX on the same surface used to bind to p53 have been previously developed and shown to be promising.

Here we engineered optogenetic control of the PMI peptide using LEXY, a light-inducible nuclear protein export system previously developed by us. LEXY enables dictating with blue light the

localization of any peptide or protein fused to it, in this case PMI and mCherry. This strategy could be applied because we found that PMI needs to be nuclear to function (Figure 1b,c).

It was interesting to see that, in HCT166 cells, nuclearly localized PMI led to higher p21 levels than the control p53-mCherry (with or without additional NLS) (Figure 1b,c). We speculate that this may be due to some interference of the fluorescent protein with p53 transcriptional activity.

It was surprising to see that the MIP peptide, which was reported to elevate p53 levels in HCT116 cells [37], did not have any effect under our experimental conditions (Supplementary Figure S1). As we did not find in the literature other reports confirming the efficacy of this peptide in living cells beyond the work by Shihiedo and colleagues, we reckon further experiments are needed to clarify this issue.

The approach based on LEXY developed here can be used to answer biological questions regarding p53 biology, such as what is the relationship between cellular outcome and p53 levels. The advantage of using light to elevate endogenous p53 levels is that cells with precisely tuned p53 levels can be generated within a single population, making the interpretation of the results easier than when comparing cells cultured separately. Moreover, optogenetics is the best approach to answer questions regarding the effect of the position of cells within a tissue on the cellular outcome, as individual cells can be illuminated. It would be, therefore, possible to study if having higher p53 levels causes different outcomes depending on whether the cells lie in the middle or at the periphery of a tissue.

PMI-LEXY represents an alternative to the existing opto-p53 which we previously established [35]. opto-p53 is the fusion of p53 to LEXY (mCherry was included for visualization purposes). This construct is transcriptionally active and allows controlling p53 target genes by controlling directly the localization of p53, not its levels. In this case, p53-LEXY, that is, an exogenous p53 protein, is the main actuator of the response. In the case of PMI-LEXY, instead, endogenous p53 is the main actuator of the response. The approach based on PMI-LEXY mimics the natural situation, while that based on p53-LEXY suffers from the possibility that alternative pathways are triggered by the high cytoplasmic p53 levels which arise during the illumination phase.

In order to advance this technology towards medical applications, it would be necessary to employ LINuS or other optogenetic tools for which light would lead to p53 accumulation, as the final aim would be to trigger the p53 signaling pathway locally, only in malignant cells. This is not possible with LEXY, as light would have to be constantly applied everywhere else in the body to keep p53 inactive. As LINuS per se works well in HCT116 cells (data not shown), we believe control of PMI with LINuS is achievable with future optimization of the construct. However, blue light penetrates poorly into tissue, thus an approach based on red/infra-red light is highly desirable.

**Supplementary Materials:** The raw data used to generate the figures in the paper are available online as supplementary figures at www.mdpi.com/xxx/. **Figure S1**: The Mdm2-inhibitory peptide (MIP) does not lead to increased p53 levels in HCT116 cells. (a) Western blot showing the levels of p53 in HCT116 cells transfected with the indicated constructs. Mock indicates cells transfected with empty pcDNA3.1(+). β-actin was used as loading control. (b) Bar plot showing the quantification of three independent western blots with samples as in (a). Data indicate mean ± standard deviation. NS p >= 0.05, *t*-test, **Figure S2**: Raw data for the plot shown in Supplementary Figure S1. Only relevant lanes were annotated, **Figure S3**: Raw data for the plot shown in Figure 1c. Only relevant lanes were annotated, **Figure S4**: Raw data for the plot shown in Figure 3c. Only relevant lanes were annotated, **Figure S5**: Raw data for the plot shown in Figure 3c. Only relevant lanes were annotated.

**Author Contributions:** Conceptualization, P.W. and B.D.V.; Methodology, P.W. and B.D.V.; Software, P.W.; Validation, P.W.; Formal Analysis, P.W. and B.D.V.; Investigation, P.W.; Resources, B.D.V.; Data Curation, P.W.; Writing – Original Draft Preparation, P.W. and B.D.V.; Writing – Review & Editing, B.D.V.; Visualization, P.W.; Supervision, B.D.V.; Project Administration, B.D.V.; Funding Acquisition, B.D.V.

**Funding:** This work was supported by the BMBF 031L0079 grant to B.D.V.

**Acknowledgments:** We thank Albert Fabregas for support with the analysis of LINuS variants in HCT116 cells and members of the Di Ventura lab for helpful discussions.

**Conflicts of Interest:** The authors declare no conflict of interest.

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
