# Peer review of "Engineering Optogenetic Control of Endogenous p53 Protein Levels"

_applsci, doi:10.3390/app9102095_

Round 1
Reviewer 1 Report
In this manuscript, the authors developed an optogenetic approach to control endogenous p53 levels with blue light. Specifically, they controlled the nuclear localization of the Mmd2-binding PMI peptide using the light-inducible export system LEXY. In the dark, the PMI-LEXY fusion is nuclear and binds to Mdm2, consenting to p53 to accumulate and transcribe the target gene p21. Blue light exposure leads to the export of the PMI-LEXY fusion into the cytosol, thereby Mdm2 is able to degrade p53 as in the absence of the peptide. This approach may be useful to study the effect of localized p53 activation within a tissue or organ.The manuscript is well written and the study is interesting.
However, the authots have mentioned that PMI-LEXY represents an alternative to the existing opto-p53 which they previously established REF. Can author explain specifically what are the benefits and shortcomings of this approach than previous reports.
Can author explain in detail about promising practical application of this approach in study of cancer or diagnosis and treatment?
Author Response
We thank the reviewer for finding the results interesting and the manuscript well written. Below we reply to each point raised by the reviewer.
However, the authots have mentioned that PMI-LEXY represents an alternative to the existing opto-p53 which they previously established REF. Can author explain specifically what are the benefits and shortcomings of this approach than previous reports.
In the initial submission, we had written that: “The major difference between PMI-LEXY and p53-LEXY is that the first allows controlling native p53, mimicking the natural way in which it gets activated in the cell.”. The reviewer is right that this sentence is not enough to explain the difference between the two. We have now expanded on this sentence and explain in more detail the difference between the two approaches (page 8 lines 403-422): “opto-p53 is the fusion of p53 to LEXY (mCherry was included for visualization purposes). This construct is transcriptionally active and allows controlling p53 target genes by controlling directly the localization of p53, not its levels. In this case, p53-LEXY, that is, an exogenous p53 protein, is the main actuator of the response. In the case of PMI-LEXY, instead, endogenous p53 is the main actuator of the response. The approach based on PMI-LEXY mimics the natural situation, while that based on p53-LEXY suffers from the possibility that alternative pathways are triggered by the high cytoplasmic p53 levels which arise during the illumination phase.”.
We also included a sentence at the very end of the Discussion to mention one shortcoming of all approaches based on AsLOV2, that is, the fact that blue light penetrates poorly into tissue (page 9, lines 428-429): “However, blue light penetrates poorly into tissue, thus an approach based on red/infra-red light is highly desirable.”
Can author explain in detail about promising practical application of this approach in study of cancer or diagnosis and treatment?
We have now changed the sentence in the Discussion to better explain how we envision that this approach may be useful in the study of p53 (page 8, lines 395-402): “The approach based on LEXY developed here can be used to answer biological questions regarding p53 biology, such as what is the relationship between cellular outcome and p53 levels. The advantage of using light to elevate endogenous p53 levels is that cells with precisely tuned p53 levels can be generated within a single population, making the interpretation of the results easier than when comparing cells cultured separately. Moreover, optogenetics is the best approach to answer questions regarding the effect of the position of cells within a tissue on the cellular outcome, as individual cells can be illuminated. It would be, therefore, possible to study if having higher p53 levels causes different outcomes depending on whether the cells lie in the middle or at the periphery of a tissue.”.
However, we would like to point out here, as we did already in the manuscript, that we do not reckon this strategy can be applied, as it is, in the context of cancer treatment. Only a LINuS-based strategy could have some potential in treatment as light would be applied only on the body part where the cancer is and would lead to the inhibition of p53-Mdm2 interaction and, as a consequence, to elevated p53 levels and pathway activation. We did write this in the paper, at the end of the Introduction (“Even if not applicable in a therapeutic scenario in the present form – as cells with elevated p53 levels are, in this case, those left in the dark –, this approach should be useful to study the effect of having a heterogeneous p53 status within a tissue or organ”) and in the Discussion: “In order to advance this technology towards medical applications, it would be necessary to employ LINuS or other optogenetic tools for which light would lead to p53 accumulation, as the final aim would be to trigger the p53 signaling pathway locally, only in malignant cells. This is not possible with LEXY, as light would have to be constantly applied everywhere else in the body to keep p53 inactive.”
Reviewer 2 Report
The authors have provided an adequate background for their study which focused on developing an optogenetic tool to inhibit the negative interaction between Mdm2 and MdmX with p53 and thus control the levels of p53 in colon carcinoma cells. They initially tested two peptides, MIP and PMI, that were previously identified to interact with Mdm2 and MdmX, for their capability to induce p53 protein elevation in transfected HCT116 cells, yet only cells transfected with thioredoxin-PMI showed elevation of p53 protein levels. Further, they introduced NLS or NES signal in their constructs and showed that the presence of NLS results in higher levels of p29 in transfected cells compared to NES. Additionally, after removal of thioredoxin-domain, they introduced the previously developed optogenetic tools LINus and LEXY in order to control the localisation of PMD with the use of blue light. None of the three LINus variants fused to PMI showed significant nuclear accumulation of PMI after light induction but PMI fused to LEXY showed reversible accumulation of the peptide in the cytosol upon light induction.
However, although the first testing of the MIP peptide was also used further on as a negative control (figure 2a), it does not give any extra value or provide any information required to follow the narrative of this manuscript. As the authors have not investigated the reason why the result is different compared to previously published work, a suggestion would be to be used as supplementary information and Figure 1 to be moved to supplementary Figures. Using this as first main figure of this paper can attract the attention from the main point which is the optogenetic control of p53 protein levels.
Importantly, statistics have not been used throughout the manuscript. Statistics are required for the interpretation of the results shown in Figures 1b, 2c, 3b, 4e, 5c. Simple t-tests can be applied to provide statistical significance to these results. In line 179 "modest yet non-negligible" as well as in line 200 "worked very well" should refer to statistical significance, otherwise can be considered vague.
In Figure 4: statistics are needed in 4e because the bar plot does not make clear the insignificance of LINus effect (or the significance of LEXY effect) in addition to an unclear accumulation of mCherry protein in the images of 4b.
The replicates of figure 2b are missing from the supplementary material although all the other western blot replicates are included.
In section 3.2, the authors do not directly show what is the effect of the removal of Trx-domain, yet they introduce the use of Daunorubicin which is not mentioned in the Materials and Methods section or anywhere else in the manuscript. A direct comparison between trx-containing construct and non would be preferred although an explanation of the use of Daunorubicin in the Materials and Methods section as well as in the Results in addition to statistics could be enough.
In line 270 a reference is missing.
Overall, the authors have clearly stated their methods and results, the above corrections and statistical tests are needed for this paper to be accepted.
Author Response
We thank the reviewer for the thorough assessment of our manuscript and for suggesting corrections that improved it. All changes to the main text are marked in red. Below we reply to each point raised by the reviewer.
However, although the first testing of the MIP peptide was also used further on as a negative control (figure 2a), it does not give any extra value or provide any information required to follow the narrative of this manuscript. As the authors have not investigated the reason why the result is different compared to previously published work, a suggestion would be to be used as supplementary information and Figure 1 to be moved to supplementary Figures. Using this as first main figure of this paper can attract the attention from the main point which is the optogenetic control of p53 protein levels.
We agree with the reviewer that the data on MIP are per seunnecessary for the rest of the story and can be shown as supplementary figure. We had included them in the main text because we did invest considerable amount of time to figure out why our results differed from the previously published ones (tested different DNA concentrations in the transfection, different cell lines) without getting anywhere. Thus we thought that it would be fair for the community to know that this peptide does not work under all conditions.
We now moved the figure to the supplement (new Supplementary Figure S1), since the statement about the inefficacy of the peptide is present in the main text.
Importantly, statistics have not been used throughout the manuscript. Statistics are required for the interpretation of the results shown in Figures 1b, 2c, 3b, 4e, 5c. Simple t-tests can be applied to provide statistical significance to these results. In line 179 "modest yet non-negligible" as well as in line 200 "worked very well" should refer to statistical significance, otherwise can be considered vague.
In Figure 4: statistics are needed in 4e because the bar plot does not make clear the insignificance of LINus effect (or the significance of LEXY effect) in addition to an unclear accumulation of mCherry protein in the images of 4b.
We agree with the reviewer that statistical analysis is required to more confidently draw conclusions and we regret not having included it already in the first submission. We have now added in every plot the statistical significance calculated using the Students’ t-test. Moreover, we have modified the qualitative expressions “modest yet non-negligible” and “worked very well” and now write “cells expressing Trx-PMI-mCherry exhibited about 60% increase in p53 levels compared to the mock transfected cells” (page 6 lines 197-198) and “We found that the PMI-mCherry fusion worked as well as the construct based on thioredoxin” (page 6 lines 280-281), respectively.
As the blot shown in Fig.2a was done only once, we have not added a bar plot showing the quantification. However, we did quantify this blot to be quantitative in the sentence.
The replicates of figure 2b are missing from the supplementary material although all the other western blot replicates are included.
We thank the reviewer for pointing this out to us. Actually, the blots were present but were not properly annotated. We have now corrected the supplementary material in a way that all the replicates for each blot are clearly annotated.
In section 3.2, the authors do not directly show what is the effect of the removal of Trx-domain, yet they introduce the use of Daunorubicin which is not mentioned in the Materials and Methods section or anywhere else in the manuscript. A direct comparison between trx-containing construct and non would be preferred although an explanation of the use of Daunorubicin in the Materials and Methods section as well as in the Results in addition to statistics could be enough.
Regarding the direct comparison between having or not the thioredoxin scaffold: actually, we had done the experiments this way, meaning that we have compared in the same blot these samples (with and without thioredoxin). We had decided, for the sake of the flow in the results part and to have simpler figures with fewer data, to show the results as two separate figures. However, given the comment of the reviewer, we now show them together (new Fig.1), so that a direct comparison can be made and the effect of the removal of the thioredoxin scaffold on the activity of the PMI peptide can be seen (there is no statistical significance between the two).
Daunorubicin was used a positive control in these blots. This is a more realistic control compared to overexpression of p53 (which we also have as additional control), as this drug activates the endogenous p53 pathway reflecting more closely what happens in the cells when p53 gets stabilized due to the inhibition of the p53-Mdm2 interaction. We have now added a sentence about daunorubicin also in the Materials and Methods section (page 3 lines 139-141).
In line 270 a reference is missing.
We thank the reviewer for spotting this mistake. We have now added the reference.